# Proboscis Extension Response of Three *Apis mellifera* Subspecies toward Water and Sugars in Subtropical Ecosystem

**Abdulaziz S. Alqarni [1,\*]**, **Hussain Ali [2]**, **Javaid Iqbal [1]** and **Hael S. A. Raweh [1]**

1   Melittology Research Lab, Department of Plant Protection, College of Food and Agriculture Sciences, King Saud University, Riyadh 11451, Saudi Arabia
2   Entomology Section, Agricultural Research Institute, Peshawar 24330, Pakistan
\*   Correspondence: alqarni@ksu.edu.sa

**Abstract:** The proboscis extension response (PER) assay revealed the responsiveness of three sub-species of the honeybee *Apis mellifera* [*A. m. jemenitica* (*AMJ*), *A. m. carnica* (*AMC*), and *A. m. ligustica* (*AML*)] to water and different concentrations (0.00001, 0.0001, 0.001, 0.01, 0.1, 0.5, 1.0, and 1.5 M) of three sugars (fructose, glucose, and sucrose) during the summer and fall seasons. The tested bee subspecies showed significantly different PERs to sugar types across the seasons. The water responsiveness of *AMJ*, a native bee subspecies, was significantly lower than that of *AMC* and *AML*, which showed an equally higher water response in both seasons. During the summer season, *AMJ* and *AMC* were equally responsive to each sugar type at all tested concentrations. *AML* was relatively less responsive to glucose at 0.001, 0.001, 0.01, 0.1, 0.5, and 1.0 M than to fructose and sucrose during the summer season. During the fall season, *AMJ* was equally responsive to glucose and sucrose at all tested concentrations but showed a significantly different response between fructose and sucrose at 0.001, 0.01, 0.1, 0.5, and 1.0 M concentrations. The PER of *AMJ* to fructose was lower than that of glucose and sucrose. *AMC* was equally responsive to all tested sugars at all concentrations, and *AML* showed a differential response between glucose and sucrose at different concentrations during the fall season. The inter-specific species comparisons revealed that all tested subspecies were equally responsive to fructose at all tested concentrations, and *AMJ* was more responsive to glucose and sucrose than *AMC* and *AML* during both seasons. *AMC* and *AML* showed no differences in PER to glucose and sucrose in either season. The *AMJ*, *AMC*, and *AML* nectar and pollen foragers showed no significant differences in PER to glucose and sucrose. The *AMC* nectar foragers were highly responsive to sucrose than pollen foragers at higher sucrose concentrations (0.1, 0.5, 1.0, and 1.5 M). The *AML* (nectar forager vs. pollen forgers) showed identical PER to sucrose and glucose but a higher response of nectar foragers to high glucose concentrations (0.5, 1.0, and 1.5 M) than pollen foragers. For water responsiveness, *AMJ* nectar and pollen foragers showed similar PER to water, whereas *AMC* and *AML* pollen foragers were significantly more responsive to water than nectar foragers.

**Keywords:** proboscis behavioral reflex; honey bee subspecies; antennal stimulation; nectar sugars; foraging bees; environmental stressors; subtropical conditions; Saudi native bee

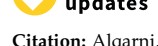



## 1. Introduction

The honeybee (*Apis mellifera* Linnaeus, 1758) is one of the most important and dominant crop pollinators in the world. It has the ability to persist in distinct climates, such as tropical, temperate and arid ecosystems all over the globe [1–3]. Based on modern taxonomic patterns, 33 subspecies of *A. mellifera* L. are recorded in different regions of the world, such as Europe, Africa, western Asia, and the Middle East [4]. The success of *A. mellifera* L. is due to its ability to adapt to various environments with diverse physiological stresses [5,6]. The commercialization of beekeeping has likely increased the opportunities for the adaptability of different honey bee species and subspecies outside of their typical geographic regions to widely dispersed new regions of the world [5,7]. The honey bee, *A.*

*mellifera* likely originated from Asia and expanded broadly into Africa and Europe [8,9]. The introduction of exotic honey bee species or subspecies in a region forces the bees to adapt to the physiological challenges of a new climate. Therefore, it is important to study the development of behavioral and physiological adaptations in honey bees.

Honey bees visit flowers to collect nectar and pollen. Bee foragers provide an opportunity to investigate the factors driving responses to environmental stressors. The genotype of foragers of different species and subspecies of honey bees is closely related to their performance in nectar and pollen collection during foraging [10,11]. Foraging decisions are flexible in foragers and are made according to the prevailing environmental and other stress conditions, including the level of stress experienced by individual foragers and the colony [12,13]. During high temperatures in summer (a type of thermal stress), foragers try to collect low concentrations of nectar and high amounts of water to maintain the environment of colonies. In addition, at higher ambient body temperature, the bees inside the colony evaporate water from their mouthparts for evaporative cooling [14]. Thus, bees have the ability to adapt to stressful fluctuating weather conditions [15].

Honey bees have an olfactory mechanism to collect nectar and pollen using their proboscis [16]. The floral scent (nectar) signal from flowers is crucial for foraging and can affect foraging activities [17–19]. Nectar is an important source of energy and consists of three major sugars, i.e., glucose, sucrose, and fructose [20–22]. The natural reflex in honey bees to nectar sugars can be monitored through the proboscis extension response (PER) by eliciting the proboscis of honey bees in response to antennal stimulation with sugar [17,23]. PER is a natural analog to the foraging of bees in the field, where honey bees use their proboscis response to perceive and collect nectar from flowers [24]. We utilized PER to test the sugar response of foragers to water and different sugar types, and this response was referred to as the sugar and water responsiveness, respectively [13,25].

PER has been widely used to establish the response thresholds of honey bees to different sugars [26]. It also reflects the nutritional conditions of the bees and their colonies [10,13]. PER is very sensitive to the season, genotype, age, and feeding status of foragers [13,27–29].

In Saudi Arabia, three subspecies of *Apis mellifera* Linnaeus, 1758 are being domesticated [1], including one native (*Apis mellifera jemenitica* Ruttner, 1976) and two exotic (*Apis mellifera carnica* Pollmann, 1879 and *Apis mellifera ligustica* Spinola, 1806) bees [30]. The climate of Saudi Arabia is hot and arid with limited rainfall. The summer is very hot, during which temperatures may exceed 45 °C [17,31]. The native bee (*A. m. jemenitica*) is well adapted and widely utilized for honey production and pollination throughout the region [32]. The exotic bees (*A. m. carnica* and *A. m. ligustica*) are imported annually and commonly distributed throughout Saudi Arabia to boost honey production. However, these exotic bees have to tolerate harsh environmental conditions of hot temperature and low humidity [1].

It was hypothesized that the differences in the genotype (subspecies) of honey bee foragers and the harsh weather conditions may have a significant impact on their foraging performance. Thus, the present study investigated the behavioral response of different honey bee subspecies regarding their responsiveness to water and different types of sugars (monosaccharides: fructose and glucose; disaccharide: sucrose), which are commonly present in floral nectar. The response to different sugar types was also determined within each subspecies.

We also compared pollen and nectar foragers within each targeted subspecies to investigate differences in their responsiveness that might reflect their physiological adaptation to local conditions. This study will provide significant knowledge to beekeepers to understand the preferential response of honey bee subspecies to sugar types, which can be used to improve the planting of suitable crops to potentially increase honey production.

## 2. Results

The response of *AMJ*, *AMC*, and *AML* forager bees to serial concentrations of three sugars (fructose, glucose, and sucrose) was investigated during the summer and fall seasons

using PER. The specificity index (SI) indicates the actual response (proportion PER) of the honey bee to sugar; calculated by subtracting the water response from the sugar response for each concentration.

### 2.1. Responsiveness of Bees (Summer Season)

2.1.1. Responsiveness of a Single Subspecies to Sugars

*AMJ* (Figure 1A) and *AMC* (Figure 1B) exhibited a similar pattern of responsiveness to fructose, glucose, and sucrose at all tested concentrations. Therefore, *AMJ* and *AMC* were equally responsive to all tested sugars and did not show any significant differences in their preferential response. *AML* showed a relatively lower response to different concentrations of glucose than fructose and sucrose (Figure 1C).

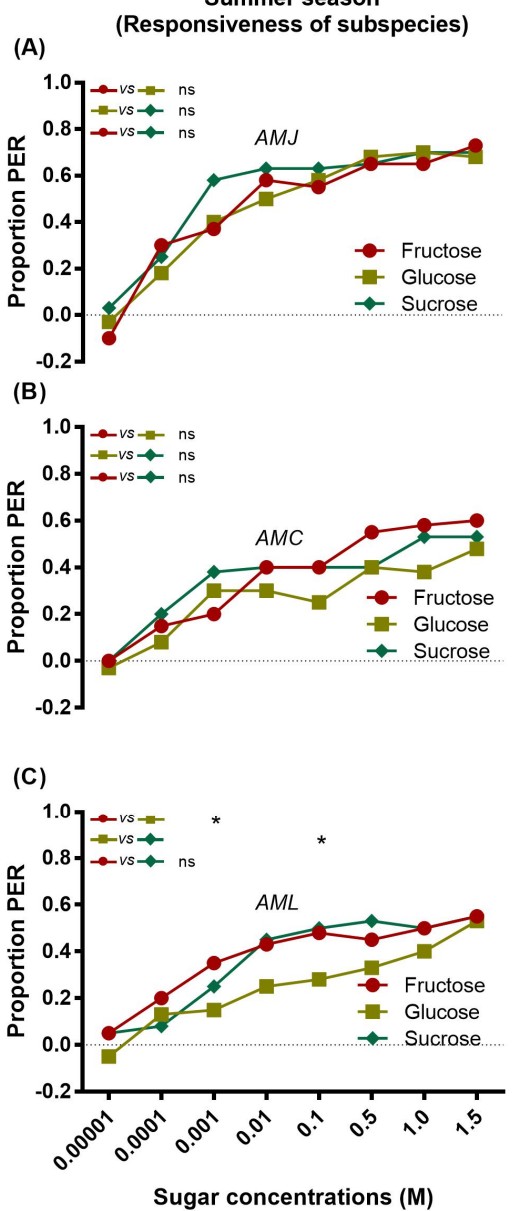

**Figure 1.** Sugar responsiveness of single honey bee subspecies *AMJ* (**A**), *AMC* (**B**) and *AML* (**C**) to fructose, glucose, and sucrose during the summer season. Asterisks indicate the significant difference ($x^2$ or Fisher's exact test * $p < 0.05$) for the responses of single honey bee subspecies *AMJ* (**A**) *AMC* (**B**) and *AML* (**C**) against different concentrations of the respective sugar (fructose vs. glucose, glucose vs. sucrose and fructose vs. sucrose).

### 2.1.2. Inter Subspecies Comparison of Responsiveness to Sugars

The comparison among honey bee subspecies regarding their responsiveness to fructose indicated no significant differences (*AMJ* vs. *AMC*, *AMC* vs. *AML*, and *AMJ* vs. *AML*) (Figure 2A). *AMJ* was highly responsive to certain concentrations of glucose (Figure 2B) and sucrose (Figure 2C) compared to *AMC* and *AML*. *AMC* and *AML* did not show any significant variations in their response to glucose and sucrose (Figure 2B,C). There was a significant difference between *AMJ* and *AML* in their responsiveness to glucose at 0.001, 0.01, 0.1, 0.5, and 1.0 M (Figure 2B) and to sucrose at 0.001 M (Figure 2C).

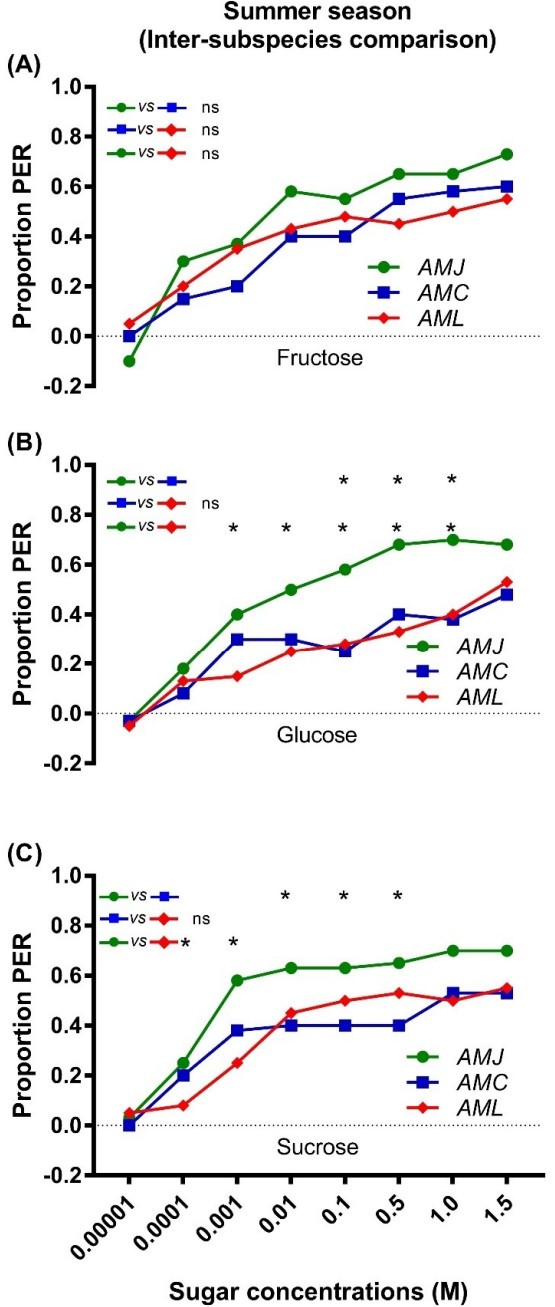

**Figure 2.** Comparison of the response to fructose (**A**), glucose (**B**) and sucrose (**C**) among honey bee subspecies during the summer season. Asterisks indicate the significant difference ($x^2$ or Fisher's exact test * $p < 0.05$) between the responses of honey bee subspecies (*AMJ* vs. *AMC*, *AMC* vs. *AML* and *AMJ* vs. *AML*) to different concentrations of each sugar type.

2.1.3. Water Responsiveness during the Summer Season

*AMJ* showed a significantly lower PER toward water than *AMC* and *AML*, which had a similar pattern of PER during the summer season (Figure 3).

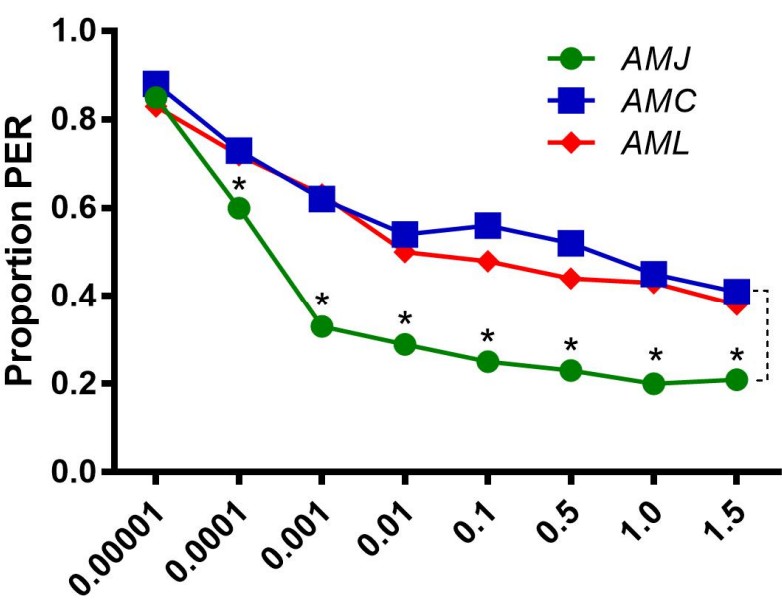

**Figure 3.** Water responsiveness, i.e., response of bees to water tested prior to the PER test to each sugar concentration. Asterisks indicate the significant difference ($x^2$ or Fisher's exact test * $p < 0.05$) among the water responses of honey bee subspecies (*AMJ*, *AMC* and *AML*).

*2.2. Responsiveness of Bees (Fall Season)*

2.2.1. Responsiveness of a Single Subspecies to Sugars

The data revealed that *AMJ* exhibited significantly different responses to sucrose and fructose at concentrations of 0.001, 0.01, 0.1, 0.5, and 1.0 M (Figure 4A), while *AMC* showed similar responsiveness to all three sugars at all tested concentrations (Figure 4B). *AML* showed a significantly different response to glucose and sucrose (Figure 4C).

2.2.2. Inter Subspecies Comparison of Responsiveness to Sugars

The comparison among honey bee subspecies regarding their responsiveness to fructose indicated no significant differences (*AMJ* vs. *AMC*, *AMC* vs. *AML*, and *AMJ* vs. *AML*) (Figure 5A). *AMJ* was highly responsive to certain concentrations of glucose (Figure 5B) and sucrose (Figure 5C) compared to *AML* and *AMC*, respectively. The exotic *AMC* and *AML* did not show any significant variations in their response to fructose, glucose, and sucrose (Figure 5ABC). Significant differences were found between *AMJ* and *AML* in their responsiveness to glucose at 0.001, 0.1, 0.5, 1.0, and 1.5 M (Figure 5B) and to sucrose at 0.01 and 0.5 M (Figure 5C). Moreover, *AMJ* and *AMC* showed significant differences in their responses to sucrose at 0.001, 0.01, 0.1, 0.5, 1.0, and 1.5 M (Figure 5C).

2.2.3. Water Responsiveness during the Fall Season

*AMJ* showed a significantly lower PER to water than *AMC* and *AML*, which showed no difference in PER to water (Figure 6). The pattern of PER to water during the fall season (Figure 6) was identical to the pattern of PER during the summer season (Figure 3).

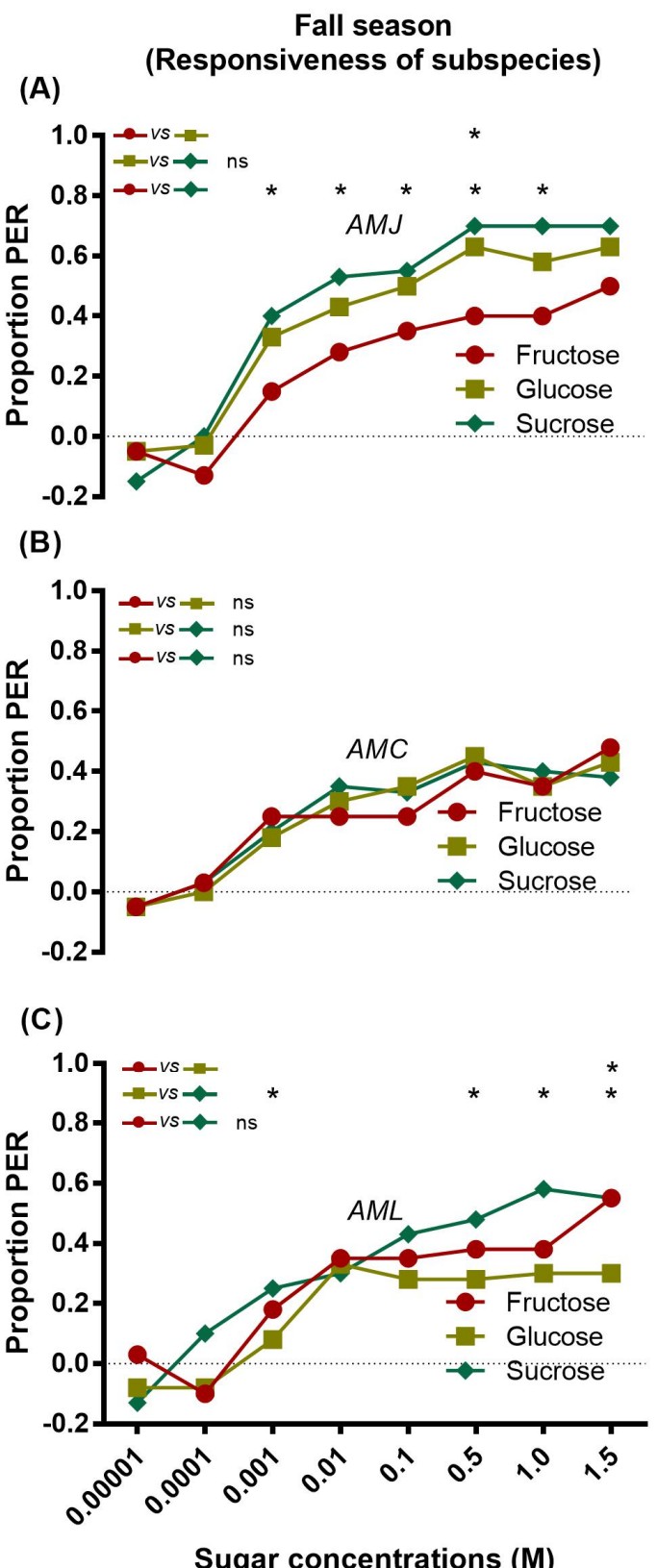

**Figure 4.** Sugar responsiveness of single honey bee subspecies *AMJ* (**A**), *AMC* (**B**) and *AML* (**C**) to fructose, glucose, and sucrose during the fall season. Asterisks indicate the significant difference ($x^2$ or Fisher's exact test * $p < 0.05$) for the responses of single honey bee subspecies *AMJ* (**A**) *AMC* (**B**) and *AML* (**C**) against different concentrations of the respective sugar (fructose vs. glucose, glucose vs. sucrose and fructose vs. sucrose).

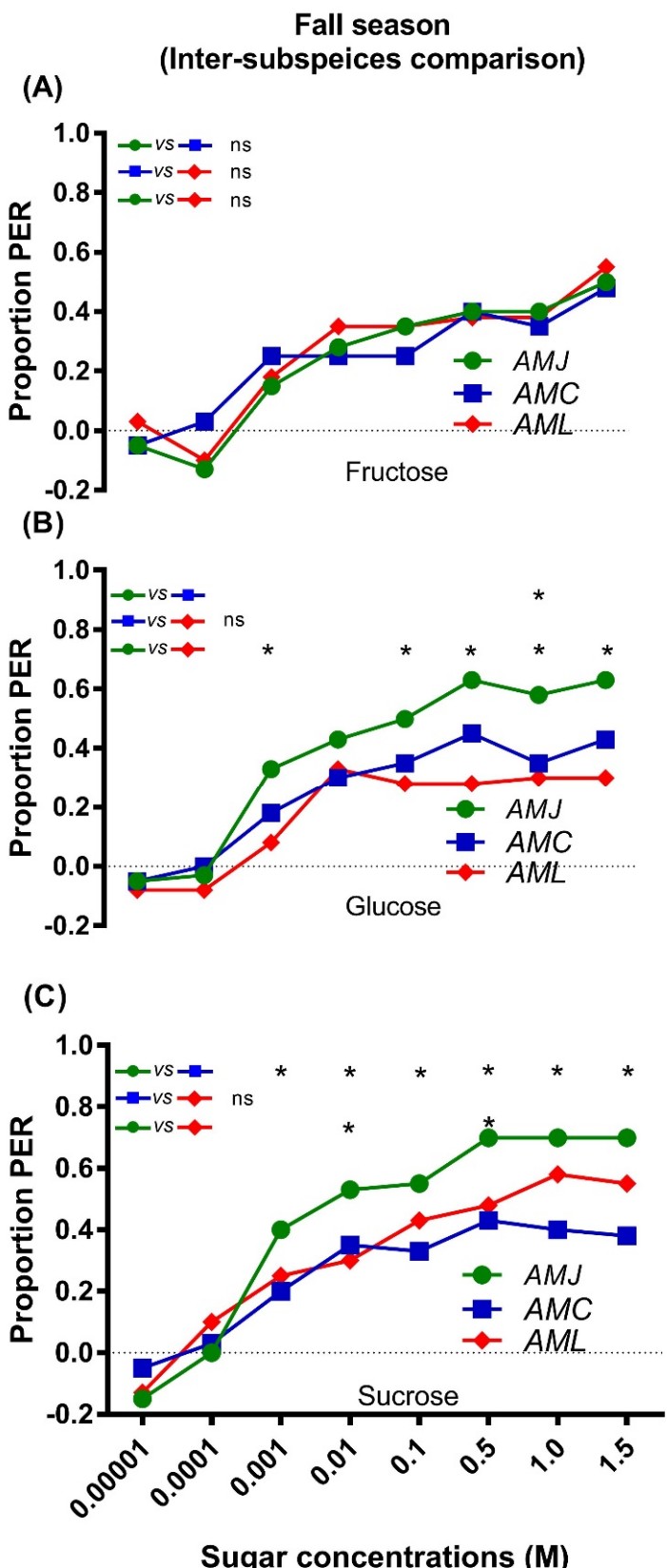

**Figure 5.** Comparison of the response to fructose (**A**), glucose (**B**) and sucrose (**C**) among honey bee subspecies during the fall season. Asterisks indicate the significant difference ($x^2$ or Fisher's exact test * $p < 0.05$) between the responses of honey bee subspecies (*AMJ* vs. *AMC*, *AMC* vs. *AML* and *AMJ* vs. *AML*) to different concentrations of each sugar type.

### 2.3. PER of Nectar vs. Pollen Foragers during the Fall Season

2.3.1. Sugar Responsiveness

The responsiveness of *AMJ*, *AMC*, and *AML* nectar and pollen foragers to glucose and sucrose were compared. The data revealed that the native *AMJ* foragers (nectar vs. pollen) showed a similar PER toward glucose (Figure 7A) and sucrose (Figure 7B), while *AMC* foragers (nectar vs. pollen) showed a similar PER toward glucose only (Figure 7C). As for the response toward sucrose, *AMC* pollen foragers were relatively less responsive to sucrose and *AMC* nectar foragers were highly responsive to higher concentrations of sucrose (0.1, 0.5, 1.0, and 1.5 M) compared to pollen foragers (Figure 7D). The PER of *AML* foragers (nectar vs. pollen) to sucrose was identical (Figure 7F). However, nectar foragers showed a higher response to high concentrations of glucose (0.5, 1.0, and 1.5 M) than pollen foragers (Figure 7E).

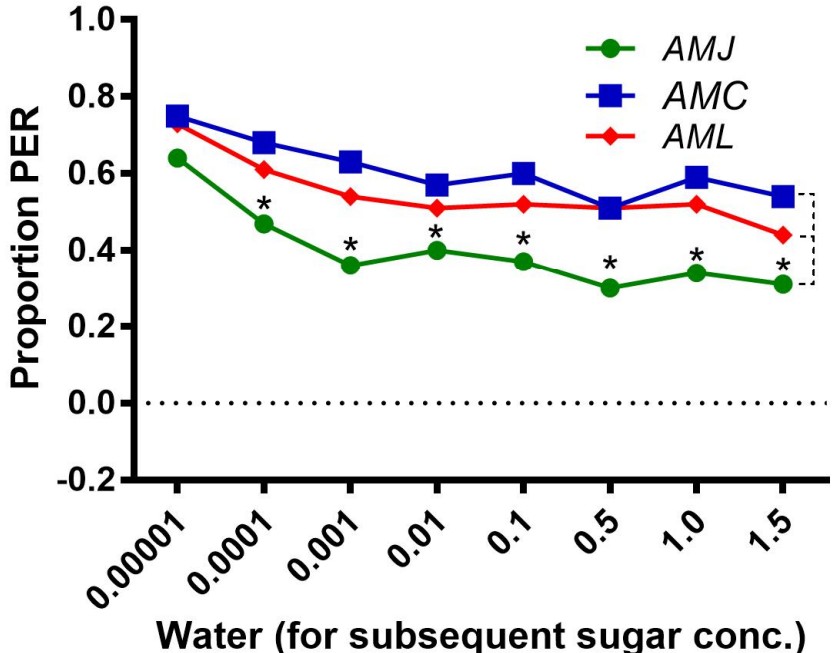

**Figure 6.** Water responsiveness, i.e., response of bees to water tested prior to the PER test to each sugar concentration. Asterisks indicate the significant difference ($x^2$ or Fisher's exact test * $p < 0.05$) among the water responses of honey bee subspecies (*AMJ*, *AMC* and *AML*).

2.3.2. Water Responsiveness

*AMJ* foragers (nectar vs. pollen) showed a similar PER toward water except at a few points with significant differences (Figure 8A). Pollen foragers of *AMC* and *AML* exhibited significantly higher PER toward water than nectar foragers, respectively (Figures 8B and 8C, respectively).

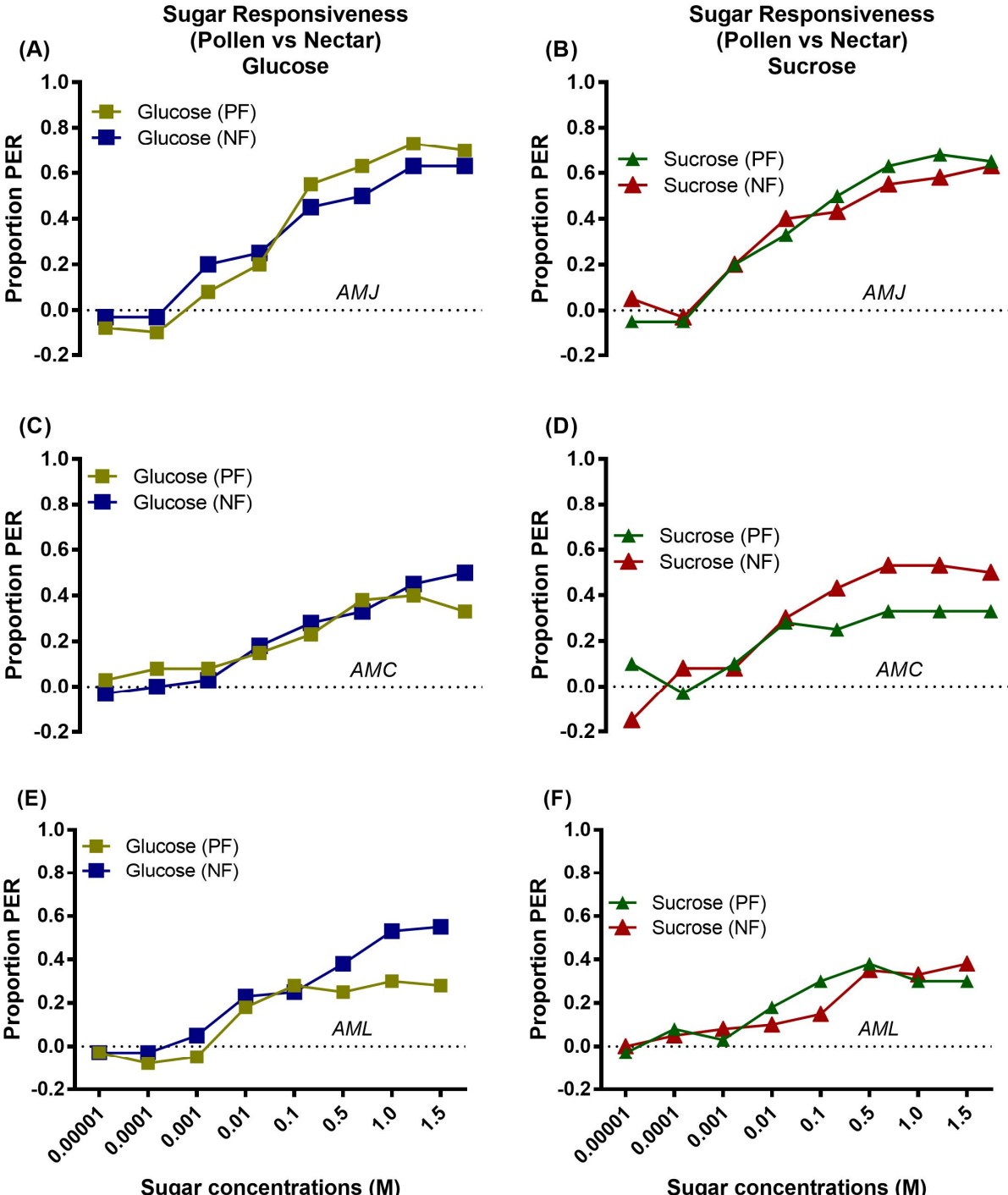

**Figure 7.** Comparison of sugar responsiveness to glucose and sucrose between pollen and nectar foragers of a single honey bee subspecies *AMJ* (**A**: glucose & **B**: sucrose), *AMC* (**C**: glucose & **D**: sucrose) and *AML* (**E**: glucose & **F**: sucrose) during the fall season. In each figure, asterisks indicate the significant difference ($x^2$ or Fisher's exact test) between the responses of pollen and nectar foragers of individual honey bee subspecies to different concentrations of glucose or sucrose.

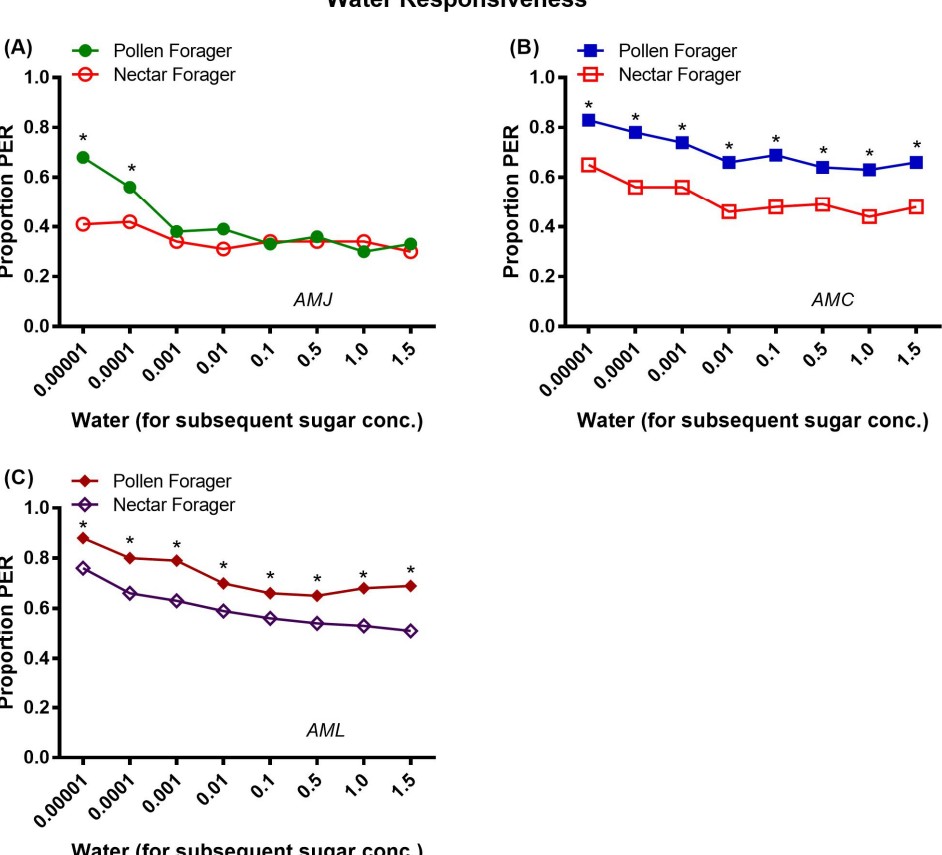

**Figure 8.** Water responsiveness, i.e., response of pollen and nectar foragers of a single honey bee subspecies *AMJ* (**A**), *AMC* (**B**) and *AML* (**C**) to water tested prior to the PER test to each sugar concentration during the fall season. Asterisks indicate the significant difference ($x^2$ or Fisher's exact test * $p < 0.05$) between the water responses of pollen and nectar foragers of individual honey bee subspecies (*AMJ*, *AMC* and *AML*).

## 3. Discussion

The three *Apis* subspecies (*AMJ*, *AMC,* and *AML*) showed differential PERs to different concentrations of three sugar types (fructose, glucose, and sucrose) across seasons. During the summer season, *AMJ* and *AMC* were equally responsive to all tested sugars at all tested concentrations, while *AML* showed variable responses to the tested sugars, and its PER to glucose was lower than that to fructose and sucrose during the summer season. During the fall season, *AMJ* and *AML* showed equal PER between glucose and sucrose but significantly different responses between fructose and sucrose, whereas *AMC* was equally responsive to all tested sugars.

The inter-subspecies comparisons revealed that all tested subspecies (*AMJ*, *AMC,* and *AML*) showed similar PER toward fructose, but *AMJ* was more responsive to glucose and sucrose than *AMC* and *AML* during both seasons. Regarding water responsiveness, *AMJ* was less responsive to water than *AMC* and *AML*, which showed higher PER toward water during both seasons.

Our data demonstrated that all tested honey bee subspecies were responsive even to lower concentrations of the tested sugars during summer, which is in partial agreement with the results of Pankiw and Page [26] who tested the PER of bees to sucrose only and found high PER to the lower sucrose concentration. However, the present study investigated and compared the response of bees to three sugar types (fructose, glucose, and sucrose).

The high-sucrose-response foragers return from the field with less concentrated nectar, presumably because they have a lower threshold to sugars compared to low-sucrose-response foragers. Pankiw et al. [33] reported that the PER of honey bees changes with

the concentration of sucrose present in the collected nectar. During summer, the weather conditions in central Saudi Arabia are quite stressful for bees [1,34]. This might lead to high PER to lower sugar concentrations due to high colony needs for water and energy. Our data showed that the imported bee subspecies (*AMC* and *AML*) showed high PER toward water in the summer season compared to the indigenous subspecies (*AMJ*), which implies the native bees' higher tolerance to harsh environmental stresses compared to the imported bees. Alqarni [34] reported significantly lower weight loss in *AMJ* workers compared to exotic *AMC* and *AML* workers after two hours' exposure in shaded cages during the summer season.

In the fall season, the response of the tested bee subspecies to lower sugar concentrations was lower than that in summer. This shows that the sensitivity to sugar varies with season, which is in line with previous studies [10,27,29,33] that reported vulnerable PER due to changes in feeding, age, foraging ability, and social pheromones released by queens and larvae. This variation provides information about the incoming resources that affect the foraging and recruitment behavior of colonies. The fluctuations in PER in summer and fall may also be related to the nectar availability and sugar concentration during the season. There are fewer flowers in the summer season in the area around Riyadh than in the fall season. The high temperature and low humidity can affect the foraging of both native and imported bees at noon time when flowers of *Acacia gerrardii* and *Ziziphus nummularia* were at their maximal phase of nectar secretion during the summer and fall seasons, respectively [35].

*Nectar and Pollen Foragers*

In the present study, the nectar and pollen foragers of the three subspecies showed similar PER toward glucose and sucrose. The PER of *AMC* nectar foragers toward sucrose was higher but non significantly different than that of *AMC* pollen foragers. Floral sources also affect the PER to sucrose in nectar and pollen foragers. Pollen foragers were more responsive to lower sucrose concentrations, while nectar foragers preferred high concentrations [10,13,29]. Our results indicated a higher PER of *AMC* nectar foragers than pollen foragers to high concentrations of sucrose, which is partially in accordance with the previously cited studies. In contrast, we did not find any significant difference between the PER of *AMJ* nectar and pollen foragers to sucrose and glucose during the fall season. *AML* also showed no difference in the PER to sucrose but a higher response of nectar foragers than pollen foragers to high concentrations of glucose.

Regarding water responsiveness, our data showed that *AMJ* foragers (nectar vs. pollen) showed identical PER toward water except for a few points with significant differences. *AMJ* is a native, well-adapted bee with high survival rates and the ability to tolerate the harsh weather of Saudi Arabia [34]. These indigenous bees have undergone some physiological changes in heat shock protein (HSP) expression for handling high temperature [36]. Our data showed that *AMC* and *AML* foragers (nectar vs. pollen) exhibited significant differences in PER toward water. Pollen foragers of *AMC* and *AML* were more responsive to water than nectar foragers. The *AMC* nectar foragers were more responsive to high sucrose concentrations than pollen foragers. This is in partial agreement with a previous study that reported that *AMC* pollen foragers were very responsive to water and sucrose throughout the foraging season [13]. The response to sugars is directly associated with the decision to collect nectar, pollen, or water [11,13]

Season and genotype can also affect PER in honey bees [10,13,37]. This was demonstrated in our experiments where the PER of the tested bee subspecies (*AMJ, AMC,* and *AML*) was not similar and significantly varied between seasons.

Further detailed studies throughout the year should be carried out to observe the fluctuations in PER of indigenous and imported honey bee subspecies to sugars and water in relation to nectar availability and sugar concentration.

## 4. Materials and Methods

### 4.1. Honey bee Subspecies

Three commonly domesticated subspecies of *Apis mellifera* (*A. m. jemenitica*, *A. m. carnica*, and *A. m. ligustica*) in Saudi Arabia were used to test their PER behavioral responsiveness. The PER phenomenon was used to elicit the reflexive response of forager bees to water and different concentrations of three sugars (fructose, glucose, and sucrose) which are commonly present in floral nectar [23,25,38,39]. Colonies of *A. m. jemenitica* (*AMJ*), the native bees of Saudi Arabia, and two exotic bee subspecies, *A. m. carnica* (*AMC*) and *A. m. ligustica* (*AML*) were reared in the apiary of Dirab agricultural research station, 40 km south of Riyadh city, Saudi Arabia. *AMJ* was obtained from a native bee stock maintained at the same KSU Dirab research station. *AMC* and *AML* colonies were reared from queens of purebred lines obtained from certified beekeepers. All bee colonies were infection free and maintained equally as per standard beekeeping methods. All summer and fall trials were conducted at room temperature ($25 \pm 5$ °C) during 2013.

### 4.2. Sample Collection and Behavioral Assay

The outgoing nectar and incoming pollen foragers were collected early from the entrances of the hives and individually placed in small glass vials for their respective experiments. The glass vials had small holes on the lid to maintain proper air exchange for the bees. The bees were immediately brought into the laboratory in a dark container, immobilized in an ice water bath for 3–5 min, and harnessed with tape to small harnessing tubes made from plastic straws [13,23,25,40]. The harnessed bees were allowed to acclimatize for at least 10 min, fed a small droplet of 0.5 M sucrose solution, and left for 2 h on the bench top at room temperature to normalize the activity of bees prior to the responsiveness test (Figure 9). The weather data, namely temperature and relative humidity, were also recorded. All behavioral trials were executed at room temperature ($25 \pm 5$ °C) during the early summer (June 2013: max. temperature 40 °C, min. temperature 27 °C) and fall (Nov 2013: max. temperature 25 °C, min. temperature 19 °C). The comparative experiments between pollen and nectar foragers were performed during fall Nov 2013 (max. temperature 25 °C, min. temperature 15 °C).

### 4.3. Behavioral Tests for Bees' Responsiveness

The PER of five bees from each subspecies to different sugar types (fructose, glucose, and sucrose) was tested daily during two seasons (summer and fall, 2013). This procedure was repeated over multiple days until a total of 40 honey bees/subspecies/sugar types were tested. The harnessed bees of each subspecies were presented with sequential serial concentrations (0.00001, 0.0001, 0.001, 0.01, 0.1, 0.5, 1.0, and 1.5 M) of fructose, glucose, and sucrose. Each bee was presented with only a single concentration of sugar in a test. A toothpick was dipped in one concentration of sugar (fructose/glucose/sucrose) or in water, and then, this immersed toothpick was touched to the antenna of harnessed bees for antennal stimulation. The bees were not allowed to feed on the immersed toothpick for sugar/water during the responsiveness test. When the bee showed its response to sugar or water by eliciting its proboscis, this was recorded as 1 for active PER response, and 0 was recorded if no response was elicited by the tested bee [22,24,25]. The PER revealed the proportion of individuals responding to the tested concentrations of sugars or water. In the comparison trials between pollen and nectar foragers, two distinct sugar types, i.e., glucose (monosaccharide) and sucrose (disaccharide), were used for the responsiveness test.

The responsiveness of bees to water was tested 3 min prior to each sugar concentration test. Distilled water was used to stimulate the bee antennae in a similar way as that used for the sugar test. This allowed us to compare the water response of each bee with the sugar response of the same bee in the preceding trial.

The specificity index (SI) was calculated by subtracting the water response of each bee from its subsequent sugar response, and the SI corresponds to the exclusive response

of bees to sugar [22]. The mean SI values of bees to each serial sugar concentration were calculated and presented in the graph as proportion PER of tested bees.

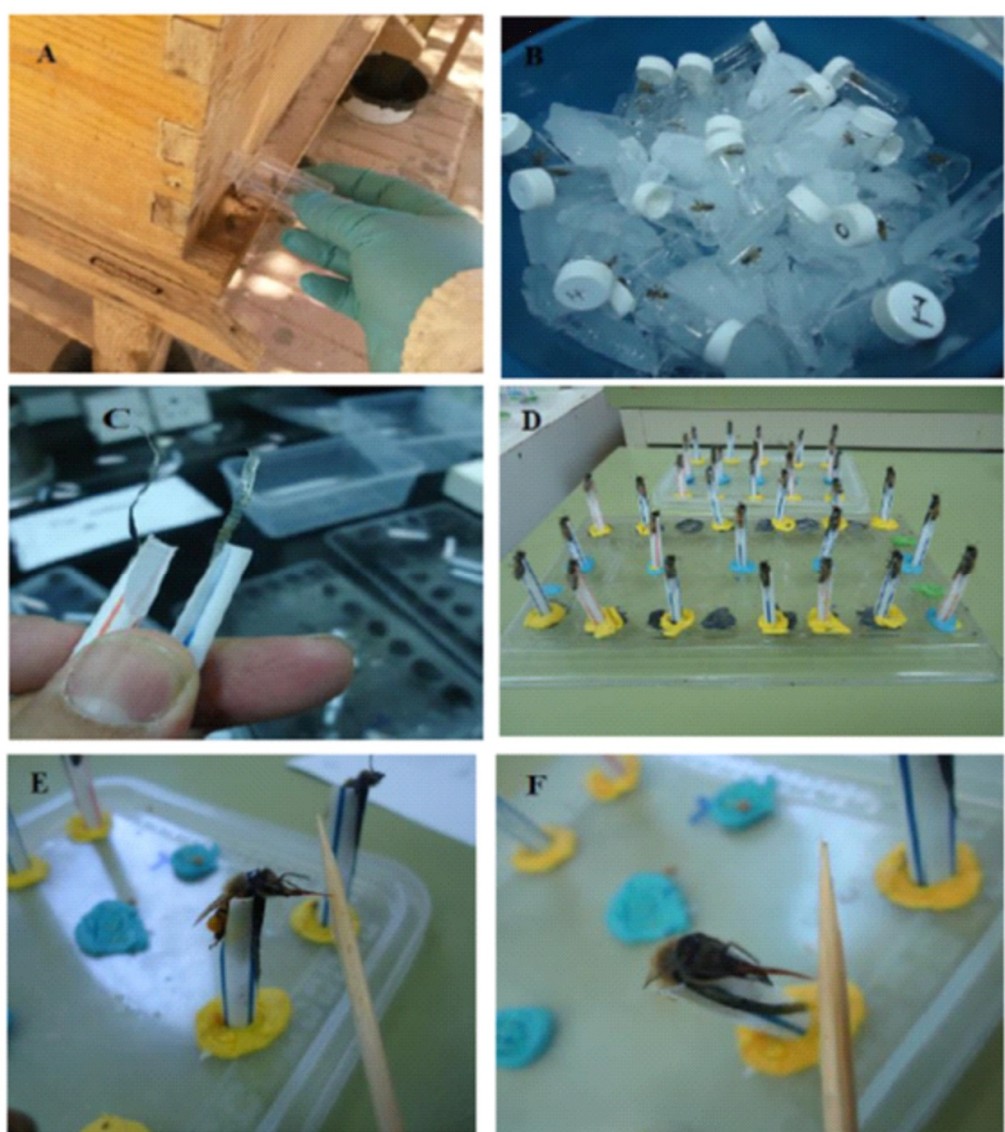

**Figure 9.** Experimental methodology. (**A**) Collection of bees; (**B**) vials on ice; (**C**) harnessing tubes (made from straws); (**D**) different subspecies after harnessing; (**E**) feeding bees before testing; (**F**) proboscis extension response testing.

### 4.4. Meteorological Data

The outdoor temperature reached an average high of 40 °C and a low of 27 °C during the summer season (May–June), and the relative humidity was 7–12% during the experimental period in the summer season. The days were mostly clear, sunny, with slight winds. During the fall season, the outdoor temperature reached an average high of 25 °C and a low of 19 °C, with 19–36% RH and mostly sunny and clear days.

### 4.5. Statistical Analysis

The statistical analyses of treatment group differences were performed using the non-parametric Kruskal–Wallis test (one-way analysis of variance). The means were compared using Dunn's multiple comparison test ($p < 0.05$) in GraphPad Prism 7. The Mann–Whitney test ($p < 0.05$) was used to analyze and compare the water responsiveness data among different bee subspecies. The response to sugar and water at each signal concentration

was compared using Pearson's nonparametric chi-square test or Fisher's exact test of proportions ($p < 0.05$).

**5. Conclusions**

Season and bee subspecies (*AMJ, AMC,* and *AML*) showed significant effects on the PER of honey bees toward different sugars (fructose, glucose, and sucrose). These differences may represent physiological adaptations to local environmental conditions. The native bees (*AMJ*) were less responsive to water than exotic bees (*AMC* and *AML*) which showed equally high responses to water in the summer and winter seasons. The water responsiveness of *AMJ* displayed patterns that are consistent with stress tolerance. *AMJ* and *AMC* were equally responsive to the tested sugars at all concentrations than *AML* which had variable responses during summer. *AMJ* was equally responsive to glucose and sucrose but had a significantly variable response between fructose and sucrose at specific concentrations during the fall season. *AMC* was equally responsive to all tested sugars at all concentrations but *AML* had a differential response to tested sugars at specific concentrations during the fall season. All tested bee subspecies were equally responsive to fructose. During both seasons, *AMJ* was more responsive to glucose and sucrose than *AMC* and *AML* which showed similar PER to glucose and sucrose. No differences between nectar and pollen forager's PER to glucose and sucrose were found in all tested bee subspecies. The nectar and pollen foragers of *AMJ* were equally responsive to water but showed significant differences in the case of *AMC* and *AML*.

**Author Contributions:** Conceptualization, A.S.A., H.A. and J.I.; methodology, A.S.A., H.A. and J.I.; validation, A.S.A. and J.I.; formal analysis, A.S.A., H.A. and J.I.; investigation, H.A., H.S.A.R. and J.I.; resources, A.S.A.; writing—original draft preparation, H.A, H.S.A.R. and J.I.; writing—review and editing, A.S.A. and J.I.; visualization, A.S.A.; supervision, A.S.A.; project administration, A.S.A. and J.I.; funding acquisition, A.S.A. All authors have read and agreed to the published version of the manuscript.

**Funding:** This research was funded by the Deputyship for Research & Innovation, Ministry of Education through the project no. IFKSURG-2-383.

**Data Availability Statement:** Data available on request due to privacy restrictions.

**Acknowledgments:** The authors extend their appreciation to the Deputyship for Research & Innovation, Ministry of Education in Saudi Arabia for funding this research work through the project no. (IFKSURG-2-383).

**Conflicts of Interest:** The authors declare no conflict of interest.

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
