# Peer review of "Proboscis Extension Response of Three Apis mellifera Subspecies toward Water and Sugars in Subtropical Ecosystem"

_stresses, doi:10.3390/stresses3010014_

Round 1
Reviewer 1 Report
1. By assessing the proboscis extension response to sugar responsiveness experiments in honeybees, the availability of specific pollen and nectar sources to honeybees may increase the uses of bee products. I read about similar articles published by the authors in 2021 (Proboscis behavioral response of four honey bee Apis species towards different concentrations of sucrose, glucose, and fructose), and evaluated the possibility of this study being published in this journal. There should be the following points that I hope can be explained more clearly:
2. The comparison was the relationship among the subspecies of Apis mellifera. Because the difference in the response of physiological adaptation in subspecific level may very small compared to the species, so the source of the subspecies of Apis mellifera should be clearly defined in the experiment. Did authors use any alignment of genotypes by molecular method to confirm the subspecies of Apis mellifera or they originates from pure bred colonies?
3. It is not stated whether there is a difference in the colony status of the experimental treatments. As far as I know, under different feeding conditions, the strength of the bee colony or the bee colony that has been infected with mite or virus should affect the performance of the sugar responsiveness of the bees in the experiment.
4. Abstract performance should be enhanced
5. The format of reference in 1, 5 and 11 should be enhanced
6. Please check the figures 1b and 4bc.
Author Response
Response to reviewer's comment is attached as separate file

Reviewer 2 Report
The authors present a simply and effective protocol to assess the Proboscis extension response of three Apis mellifera subspecies toward water and sugars in subtropical ecosystem".
I believe the conclusion is too succinct. I personally cannot understand it. In opposition, the Abstract is very long and too detailed. The Conclusion and the abstract should in fact be reversed.
Please note "Apis mellifera L. subspecies" is incorrect. You should list the correct classifier for each of the three subspecies the first time they are mentioned in the text. For example apis mellifera jemenitica Ruttner, 1976.
You should only list Apis mellifera L. when speaking about western bees - in general.
I only have questions for the conclusion which again is too simple and should be changed with the abstract:
The native bees were less responsive to water and equally responsive to the tested sugars at all concentrations” - please clarify: native bees (A. m. jemenitica). The sentence seems to indicate more than one, but the only native one tested was A. m. jemenitica. Perhaps they mean equally responsive and the exotic subspecies carnica and ligustica.
“Season and bee genotype showed significant effects on the PER of honey bees toward different sugars (fructose, glucose, and sucrose)”. Please explain for each exotic and native subspecies this result.
“Bee genotype” - I think this term is not correct as the genotype was not tested. I understand authors mean subspecies, hence they should list subspecies, as they did in the rest of the paper.
Author Response
Response to reviewer's comments is attached as separate file

Round 2
Reviewer 1 Report
The author has made appropriate revisions based on comments. I agree that the text is suitable for submission to this journal and ask the editor-in-chief for follow-up evaluation.
Reviewer 2 Report
it's OK to publish